# Rare truncating variants in the sarcomeric protein titin associate with familial and early-onset atrial fibrillation

Gustav Ahlberg[1,2], Lena Refsgaard [1,2], Pia R. Lundegaard[1,2], Laura Andreasen[1,2], Mattis F. Ranthe[3], Nora Linscheid[4,5], Jonas B. Nielsen[6], Mads Melbye [3,6,7], Stig Haunsø[1], Ahmad Sajadieh[8], Lu Camp[9], Søren-Peter Olesen[2], Simon Rasmussen[10], Alicia Lundby[4,5], Patrick T. Ellinor [11,12], Anders G. Holst[1], Jesper H. Svendsen [1,6] & Morten S. Olesen[1,2]

A family history of atrial fibrillation constitutes a substantial risk of developing the disease, however, the pathogenesis of this complex disease is poorly understood. We perform whole-exome sequencing on 24 families with at least three family members diagnosed with atrial fibrillation (AF) and find that titin-truncating variants (TTNtv) are significantly enriched in these patients ($P = 1.76 \times 10^{-6}$). This finding is replicated in an independent cohort of early-onset lone AF patients ($n = 399$; odds ratio = 36.8; $P = 4.13 \times 10^{-6}$). A CRISPR/Cas9 modified zebrafish carrying a truncating variant of titin is used to investigate TTNtv effect in atrial development. We observe compromised assembly of the sarcomere in both atria and ventricle, longer PR interval, and heterozygous adult zebrafish have a higher degree of fibrosis in the atria, indicating that TTNtv are important risk factors for AF. This aligns with the early onset of the disease and adds an important dimension to the understanding of the molecular predisposition for AF.

[1] Laboratory for Molecular Cardiology, Department of Cardiology, The Heart Centre, Rigshospitalet, University Hospital of Copenhagen, Copenhagen 2100 Ø, Denmark. [2] Department of Biomedical Sciences, University of Copenhagen, Copenhagen 2200 N, Denmark. [3] Department of Epidemiology Research, Statens Serum Institute, Copenhagen 2300 S, Denmark. [4] Cardiac Proteomics Group, Department of Biomedical Sciences, Faculty of Health and Medical Sciences, University of Copenhagen, Copenhagen 2200 N, Denmark. [5] Novo Nordisk Foundation Center for Protein Research, Faculty of Health and Medical Sciences, University of Copenhagen, Copenhagen 2200 N, Denmark. [6] Department of Clinical Medicine, Faculty of Health and Medical Sciences, University of Copenhagen, Copenhagen 2200 N, Denmark. [7] Department of Medicine, Stanford University School of Medicine, Stanford 94305 CA, USA. [8] Department of Cardiology, Copenhagen University Hospital, Bispebjerg, Copenhagen 2400, Denmark. [9] The Lundbeck Foundation Centre for Applied Medical Genomics in Personalized Disease Prediction, Prevention and Care, Copenhagen 2200 N, Denmark. [10] Department of Bio and Health Informatics, Technical University of Denmark, Kgs, Lyngby 2800, Denmark. [11] Cardiovascular Research Centre, Massachusetts General Hospital, Boston 02114 MA, USA. [12] Program in Population and Medical Genetics, The Broad Institute of Harvard and MIT, Cambridge 02114 MA, USA. These authors contributed equally: Lena Refsgaard, Gustav Ahlberg, Pia R. Lundegaard.  Correspondence and requests for materials should be addressed to M.S.O. (email: mortensol@sund.ku.dk)

Atrial fibrillation (AF) is the most common cardiac arrhythmia in clinical practice. At present, >30 million people are affected worldwide[1,2]. AF is a major risk factor for stroke, heart failure, and premature death[3], making AF a major public healthcare burden[4].

Genome-wide association studies (GWAS) have identified 97 common AF susceptibility loci[5,6] and candidate gene studies have implicated rare genetic variants in a number of genes[7], primarily encoding cardiac ion channels. Recent GWAS have associated AF with genetic loci located in the proximity of genes encoding structural proteins, such as *SYNPO2L*[8] and *TTN*[5]. Furthermore, two studies[9,10] have independently identified an association between early-onset AF and a rare variant in the myosin-coding gene *MYL4*. These recent findings suggest that AF could be caused by atrial cardiomyopathy[11,12]. Identifying highly predisposing genetic variants could lead to a better understanding of the underlying pathophysiological mechanisms and provide better means for risk stratification and treatment of AF.

In the present study, we investigate the genetic background of familial and early-onset AF, aiming to identify genes involved in AF pathology. We perform whole-exome sequencing of families with three or more family members affected by AF. In these families, we find an increased burden of *TTN* mutations, which is replicated in an independent lone AF cohort. Using a zebrafish model carrying an I-band truncating mutation in *ttn.2*, we show that heterozygous mutants have a distinct sarcomere defect. Further analyses of the heart reveal an increased amount of fibrosis in mutant fish, suggesting a predisposition for arrhythmia and conduction disease. In line with these findings, electrocardiographic analysis of adult zebrafish reveal an electrophysiological defect, represented by prolonged PR interval, a well-recognized marker for increased risk of AF.

AF has traditionally been described as an electrical disease of the heart. Our current study indicates that defects in genes involved in the structural architecture of the heart also play a significant role in the development of AF.

## Results

**Study subjects.** Family cohort: Twenty-four families were eligible for inclusion (at least three participating family members had an AF diagnosis) and agreed to participate, resulting in 77 AF cases in total (Supplementary Fig. 1).

Early-onset lone AF: A total of 399 patients were recruited using the Danish National Patient Registries. Only patients with onset of disease before the age of 40 with no known AF risk factors and normal echocardiography were included. Summary characteristics of the early-onset lone AF cohort are presented in Table 1.

Two control groups were included in the study. Control group A consists of 663 individuals in an exome-sequenced subpopulation from the Inter99 study[13,14], further described in the Methods section. Control group B consists of 383 individuals without cardiac disease, who have undergone cardiac examination and Holter monitoring as part of the Copenhagen Holter Study[15].

**Inclusion of families.** With the use of the Danish National Registries[16], we identified all Danish families with three or more members diagnosed with AF. In total, 67 families with 213 AF cases were identified. These were contacted by mail with an invitation to participate in the study. We were unable to contact 20 AF cases, due to unknown addresses. Thirty-one invited individuals replied that they did not wish to participate and 49 did not reply. One hundred thirteen individuals, from 42 families, accepted the invitation. In order to be included into the cohort, at least three members, from the same family, with the diagnosis of

AF had to agree to participate, e.g., if less than three family members agreed, the family was not included. Twenty-four families fulfilled the inclusion criteria, which resulted in 77 AF cases for exome sequencing.

**Evaluation of sample quality and population structure.** The minimum call rate in AF family exomes was 98.4% and in control exomes 97.0%. Eight control exomes were excluded on the basis of a singleton rate > 4 standard deviations (SDs). A high singleton rate could imply ancestry of unmatched ethnicity. We then analyzed the population structure in our groups with Multidimensional Scaling Analyses (MDS) using unrelated 1000 genomes project samples as reference ($n = 25,455$ single-nucleotide polymorphisms (SNPs); linkage disequilibrium (LD) threshold > 0.5; minor allele frequency (MAF) > 1%, excluding non-autosomal and monomorphic variants). Three control samples were considered ethnic outliers (Supplementary Fig. 2). The fixation index ($F_{st}$) was estimated to 0.00122, between index patients and controls (samples $n = 687$, SNPs $n = 22,745$). The MDS and relationship analyses agreed with reported ethnicity and relatedness and we did not observe any batch effects (Supplementary Figs. 3, 4, Supplementary Table 1).

Samples in the replication study that did not meet quality control criteria and ethnic outliers were excluded (see Methods). The MDS analyses agreed with reported ethnicity (Supplementary Fig. 5). We included 770 ethnically matched and unrelated individuals for a subsequent burden test (lone AF cohort = 395; control group B = 375).

**Sequencing coverage.** The mean depth of coverage in control exomes was 82.1X (SD = 13.3, $n = 663$) and in AF family exomes 94.6X (SD = 11.4, $n = 77$) (excluding non-autosomal and filtered regions). Constitutively cardiac expressed *TTN* exons (percent spliced in [PSI] > 90) were considered low coverage if < 80% of samples in either the case or the control group were covered with <20X reads on average. Sixteen *TTN* exons had low coverage and

---

**Table 1 Baseline characteristics of the early-onset lone AF cohort**

| | Early-onset lone AF ($n = 399$) | Control group B ($n = 383$) |
|---|---|---|
| Sex, male ($n$ (%)) | 334 (84%) | 257 (67%) |
| Age, years (median (IQR)) | 28 (23–33)[a] | 71 (66–76)[b] |
| Height, cm (mean (SD)) | 183 (± 8.9) | 172 (± 8.8) |
| Weight, kg (mean (SD)) | 91.4 (± 49.1) | 77.4 (± 14.7) |
| BMI, kg m$^{-2}$ (mean (SD)) | 27 (± 15.5) | 26.4 (± 4.9) |
| Smoking, current or previously ($n$ (%)) | 119 (29%) | |
| Alcohol consumption per week (mean (SD)) | 6.0 (± 7.2) | |
| Hypertension ($n$ (%)) | 0 (0%) | 246 (64.4%) |
| Diabetes ($n$ (%)) | 0 (0%) | 39 (10.2%) |
| Heart failure ($n$ (%)) | 0 (0%) | 0 (0%) |
| Myocardial infarction ($n$ (%)) | 0 (0%) | 0 (0%) |
| Valvular heart disease ($n$ (%))[c] | 0 (0%) | 0 (0%) |
| Type of AF ($n$ (%)) | | |
| Paroxysmal | 246 (62%) | |
| Persistent | 126 (32%) | |
| Permanent | 23 (6%) | |
| Family history, self-reported ($n$ (%)) | 157 (39%) | |

AF: atrial fibrillation, IQR: interquartile range, SD: standard deviation
[a]AF onset age
[b]Age at enrolment
[c]Severe mitral regurgitation, aortic stenosis, tricuspid regurgitation, or ≥ mild mitral stenosis

were not included in subsequent analysis (Supplementary Fig. 6). These exons amount to ~4% of the coding bases in *TTN* cardiac expressed isoforms (PSI > 90).

In the replication study, we employed a target deep sequencing approach. The mean depth of coverage of *TTN* in the lone AF cohort and control group B was 833X (SD = 155, $n = 770$). The mean depth of coverage in the lone AF group and control group B were 877X (SD = 135, $n = 395$) and 900X (SD = 173, $n = 375$), respectively. For each sample included in the replication test, > 99% of the targeted bases within *TTN* were covered at least 20 times.

**Genetic variation.** Whole-exome sequencing analysis was performed on the 24 families in the familial AF cohort ($n = 77$) and was compared with control group A ($n = 663$). We aimed to identify rare loss-of-function (LOF) variants with an autosomal dominant disease pattern and high penetrance. Only family members with the diagnosis of AF were taken into consideration, since 40–80% of all AF episodes are asymptomatic[17,18].

In 16 out of 24 families, we identified rare co-segregating LOF variants (Supplementary Data 1). Five of the 24 families (21%) had a predicted LOF variant, i.e., stop-gain-, frameshift-, or splice site variants, in well-established cardiomyopathy disease-causing genes. Four plausible pathogenic titin-truncating variants (TTNtv) (ENST00000589042; c.6508 G > T, c.79101 T > A, c.11952 C > A, c.98300delT) and one LOF variant in *DSC2* (ENST00000280904; c.475-1 G > T) were identified among five unrelated families (Fig. 1, Table 2). All variants co-segregated with disease, however, this does not imply a monogenetic cause. Post analysis, we looked for rare missense variation in genes previously associated with familial AF (*GJA5*, *KCNA5*, *KCNE2*, *KCNJ2*, *KCNQ1*, *KCNH2*, *NPPA*, and *SCN5A*), which was absent[9].

To validate this finding, we investigated the variation of rare TTNtv in an early-onset lone AF cohort and control group B. Summary characteristics of the early-onset lone AF cohort are presented in Table 1. Sixteen rare TTNtv, distributed among 18 early-onset lone AF cases (4.7%), were identified (Supplementary Data 2). All identified variants reside in constitutive exons (PSI ≥ 90) and are incorporated into the two principal cardiac isoforms (N2BA and N2B). Among these, 11 are found in the A-band, 6 in the I-band, 4 in the M-line, and 1 in the Z-disc (Supplementary Fig. 7). All identified TTNtv, except one, were absent from the Exome Aggregation Consortium (ExAC) database (Supplementary Data 2).

**Association tests of TTNtv.** We tested the association between TTNtv and AF using familial AF cases and control cohort A. A generalized linear mixed model (GLMM) burden test with TTNtv revealed a significant association between AF and TTNtv (AF family members = 76, control group A = 663, $P = 1.76 \times 10^{-6}$). This result corresponds to a TTNtv (PSI ≥ 90) frequency of 16.7% in the familial AF cohort and 0.5% in the control population.

Seventeen rare TTNtv were identified in the early-onset lone AF cohort and no TTNtv were identified in control group B (allele count cases = 18, allele count controls = 0). Using Firth logistic regression, a significant association between TTNtv and early-onset lone AF was replicated in the early-onset lone AF cohort and control group B (odds ratio [OR] = 36.8, 95% confidence interval [CI] = 5.0–4692.5, $P = 4.13 \times 10^{-6}$).

**Clinical characteristics of TTNtv carriers.** All TTNtv carrying family members had early-onset AF (age of onset ≤ 65 years; median age of onset = 40 years) and had been examined with echocardiographic examinations and a revision of medication history (Fig. 1, Table 3, Supplementary Table 2). The clinical data

suggest that left atria (LA) dilations could be tachycardia-induced (Table 3). For example, M1023_III-3 was diagnosed with AF at the age of 25 (LA diastolic diameter = 38 mm; left ventricle ejection fraction [LVEF] = 66%). In relation to an episode of persistent AF with high ventricular rate occurring 27 years after diagnosis, the same patient developed mild LA dilation, heart failure, and a mildly dilated left ventricle (LV) (LA = 44 mm; LVEF = 25%; LV diastolic diameter = 56 mm; Table 3). Interestingly, just 2 weeks prior to the described event, a pre-planned echocardiography with normal findings was performed. Also, E003_II-2 developed severely dilated atria at the time of transition from paroxysmal to persistent AF, with normalization after rate control. This patient's latest available echocardiography (18 years after AF diagnosis) was normal, demonstrating complete remission. A genetic predisposition for structural changes in the atrial myocardium with increased arrhythmia burden therefore seems plausible (Fig. 1). Three out of 15 patients (E003_III-4, M1023_III-1, and M1039_III-3) were diagnosed with dilated cardiomyopathy (DCM) at the same time as their AF diagnosis. Two out of these three had echocardiographic remission after frequency control of their AF, indicating tachycardia-induced disease.

The median age of onset for TTNtv carriers in the early-onset lone AF cohort was 26 (Supplementary Data 2). None of the carriers had known risk factors for AF at the time of diagnosis. Echocardiographic examinations at the time of AF diagnosis or earliest possible thereafter, showed normal echocardiographic parameters for all carriers, with regard to LV dimensions, LA size, and LVEF. One patient, patient 2, had a moderately enlarged LA, but normal ventricular size and function, and the LA dilation was considered tachycardia induced. Follow-up clinical echocardiographic examinations, some performed decades after the AF diagnosis, have revealed normal heart function and ventricular sizes (a total of 200 person-years of life after AF diagnosis; median follow-up = 9 years; Supplementary Data 2). To date, none of the TTNtv carriers have been diagnosed with DCM.

**Proteomics.** Proteomics experiments were conducted to evaluate the protein expression level of titin in human atria compared with human ventricles. Titin is among the 10 most abundant proteins in the human heart evaluated by high-resolution mass spectrometry (MS) measurements (Supplementary Fig. 8a). The physiological role of titin has predominantly been investigated in the ventricles, and we did not find a statistically significant difference between the protein expression levels of titin in the atria compared with the LV (Supplementary Fig. 8b).

**Zebrafish model.** Defects in sarcomere assembly in early development would suggest that TTNtv predispose both chambers for disease, independent of long-term systemic effects. To address the molecular consequences of a TTNtv in early cardiac development, we used the zebrafish as a model. Zou et al. have previously shown that a CRISPR/Cas9 generated homozygous mutant carrying an N-terminal truncated variant of titin, *ttn.2^{sfc9}*, displayed a severe sarcomere defect[19]. The truncation is in proximity to one of the patient-specific mutations identified in the current study, and this mutant zebrafish was used to further investigate the potential effects of TTNtv. The Z-discs of WT, heterozygous, and homozygous larvae were stained with the z-disc protein, alpha-actinin, and visualized with confocal microscopy (Supplementary Figs. 9–11). We replicated the previous findings by Zoe et al., where the WT and heterozygous larvae appeared with normal Z-discs, whereas the homozygous mutants showed a severe sarcomere defect, with absent Z-discs. The homozygous variant in zebrafish is embryonic lethal, and the larvae die around 5 days

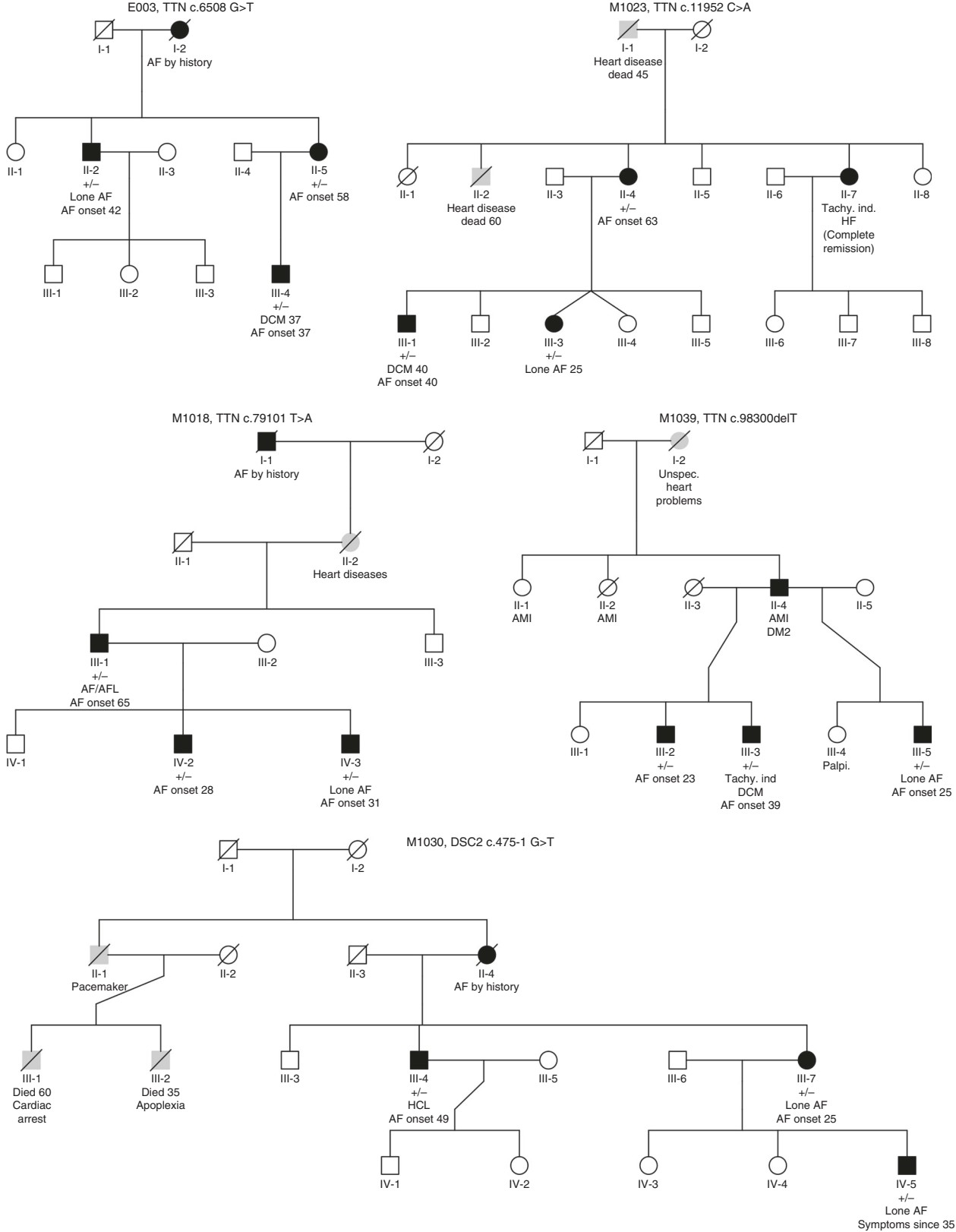

**Fig. 1** Pedigrees of the families with TTNtv and a loss-of-function variant in *DSC2*. Square: male. Circle: female; Black filled: AF affected individual; White filled: unaffected individual; Gray filled: individual with symptoms of heart disease; Diagonal line: diseased individual. Presence of mutation is indicated with + for presence and − for absence (persons with available exomes). Patient II_6 also had thyroid disease and aortic regurgitation. Tachy.Ind.: tachycardia induced, HF: heart failure, AMI: acute myocardial infarction, DM2: diabetes type 2, HCL: hypercholesterolemia, DCM: dilated cardiomyopathy

**Table 2 AF-associated loss-of-function variants in AF families**

| Family | Gene | Genomic position (hg19) | Variant | Consequence | ExAC MAF | GERP score | Exon | PSI |
|---|---|---|---|---|---|---|---|---|
| E003 | TTN | 2:179640083 | c.6508 G > T | Splice site | 0 | 5.27 | 28 | 100 |
| M1023 | TTN | 2:179606008 | c. 11952 C > A | Stop-gain | 0 | 5.87 | 49 | 100 |
| M1018 | TTN | 2:179431758 | c.79101 T > A | Stop-gain | 0 | 5.62 | 327 | 100 |
| M1039 | TTN | 2:179404492 | c.98300delT | Frameshift | $8.3 \times 10^{-6}$ | NA | 353 | 100 |
| M1030 | DSC2 | 18:28669558 | c.475-1 G > T | Splice site | 0 | 6.16 | (Intron 4) | NA |

AF: atrial fibrillation, ExAC: Exome Aggregation Consortium, GERP: genomic evolutionary rate profiling, PSI: percent spliced in

**Table 3 Clinical information on the TTNtv and DSC2 loss-of-function carriers in the AF families**

| Family | Pedigree ID | Sex | AF onset age | Age | Comorbidities/risk factors for AF[a] | ECHO, at AF diagnosis or earliest available | | | | ECHO, latest available | | | |
|---|---|---|---|---|---|---|---|---|---|---|---|---|---|
| | | | | | | LVDD (mm) | LA (mm) | LVEF (%) | Years from AF diagnosis | LVDD (mm) | LA (mm) | LVEF (%) | Years from AF diagnosis |
| E003 | II-2 | M | 42 | 62 | | N[b] | Sd[b] | 40–45[c] | 16 | N[b] | 40 | 55 | 18 |
| E003 | II-5 | F | 58 | 71 | HTN, HT | N[b] | 45 | N[b] | 4 | N[b] | Sd[b] | N[b] | 10 |
| E003 | III-4 | M | 37 | 47 | NIDCM | 67 | 57 | 15 | 0 | 70 | 56 | 30–35 | 8 |
| M1023 | II-4 | F | 63 | 71 | | N[b] | N[b] | 35[c] | 0 | 60 | 32 | 25[c] | 7 |
| M1023 | III-1 | M | 40 | 50 | NIDCM | 67–71 | 51 | 20 | 3 | 54 | 44 | 43[c] | 11 |
| M1023 | III-3 | F | 25 | 52 | | 49 | 38 | 66 | 22 | 56 | 44 | 25[c] | 27 |
| M1018 | III-1 | M | 65 | 69 | Multi[d] | Sd[b] | Sd[b] | N[b] | 0 | Mild CLVH[b] | D[b] | 60 | 3 |
| M1018 | IV-2 | M | 28 | 38 | HTN | N[b] | N[b] | N[b] | 0 | N[b] | N[b] | N[b] | 0 |
| M1018 | IV-3 | M | 31 | 40 | | N[b] | 34 | N[b] | 0 | N[b] | N[b] | 60 | 7 |
| M1039 | III-2 | M | 23 | 50 | | N[b] | N[b] | N[b] | 16 | 55 | N[b] | 60 | 24 |
| M1039 | III-3 | M | 39 | 51 | NIDCM | Sd[b] | N.A. | 30[c] | 5 | Moderat D[b] | N[b] | 50–55 | 8 |
| M1039 | III-5 | M | 25 | 37 | | N[b] | N[b] | N[b] | 9 | N[b] | N[b] | 60 | 11 |
| M1030 | III-4 | M | 49 | 63 | HCL | N[b] | D[b] | N[b] | 0 | N[b] | D[b] | N[b] | 0 |
| M1030 | III-7 | F | 50 | 72 | | Mild H | Mild D[b] | N[b] | 10 | Mild H | Mild D[b] | N[b] | 0 |
| M1030 | IV-5 | M | 41 | 46 | | 52 | 43 | N[b] | 0 | 52 | 43 | N[b] | 0 |

AF: atrial fibrillation, LVDD: left ventricular diastolic diameter, LA: left atrial diameter, LVEF(%): left ventricular ejection fraction (%), D: dilated, H: hypertrophy, HT: hyperthyroidism, HTN: hypertension, HCL: hypercholesterolemia, NIDCM: nonischemic dilated cardiomyopathy, M/F: male/female, N: normal, Sd: severely dilated, CLVH: concentric left ventricular hypertrophy, TTN: titin, TTNtv: titin-truncating variants, DSC2: desmocollin 2, N.A.: not available, ECHO: echocardiography
[a]Diagnosed before or within 12 months after the AF diagnosis
[b]Exact values not provided
[c]During AF
[d]Hypertension, stroke, aortic-, and mitral prolapse and regurgitation (biological aortic valve and mitral valve repair), dilated LV

post fertilization. All patients carrying TTNtv in this study were heterozygous adults and we therefore decided to study the adult fish. To observe the sarcomere in greater detail, we performed transmission electron microscopy (TEM) on isolated hearts from early larval stage (WT, homo- and heterozygous), adult heterozygous and WT fish. The TEM analysis revealed a sarcomere defect in both atria and ventricle of the adult and larval heterozygous mutants (Fig. 2, Supplementary Figs. 1, 2). Adult mutants showed highly disorganized sarcomeres in both chambers, with poorly defined Z-discs and absent I-bands and M-lines. Furthermore, the myofibril structure appeared loose and the sarcomere length was significantly shorter (Supplementary Figs. 1, 3, Supplementary Tables 3, 4).

Studies of electrical function using electrocardiogram (ECG) analyses showed a prolonged PR interval in the heterozygous zebrafish (Supplementary Table 5).

With the use of Sirius Red staining, we investigated the amount of fibrosis in the heterozygous *ttn.2$^{sfc9}$* adults and WT adult zebrafish. The staining revealed a higher degree of fibrosis in the atria of the mutant fish as compared with their WT siblings (Fig. 3).

## Discussion

We identified a significant enrichment of TTNtv in patients with familial AF ($P = 1.76 \times 10^{-6}$). The association was validated in an independent cohort of early-onset lone AF cases (OR = 36.8, $P = 4.13 \times 10^{-6}$). This result suggests TTNtv to be a major genetic contributor to AF, adding a substantial risk to this complex oligogenic trait. The *TTN* gene encodes the giant sarcomere protein titin that is highly expressed in all chambers of the human heart (Supplementary Fig. 8). It is well known that *TTN* dysfunction can alter ventricular cardiomyocyte structure through its involvement in multiple functions such as sarcomere assembly and passive elasticity of muscles[20]. TTNtv have previously been strongly associated with cardiomyopathies[21] and some studies have reported an increased burden of arrhythmias among DCM patients with TTNtv[22,23]. Schafer et al. showed that TTNtv encoded into cardiac exons (PSI > 15) is found in approximately 0.5% of the general population, which give rise to a continuum of phenotypes and compromises both metabolic function and cardiac performance[24]. Our result is congruent with this estimate, since 0.5% in control group A had TTNtv, whereas 16% of patients with familial AF and 4.7% of patients with early-onset lone AF had TTNtv. All rare TTNtv identified in the AF families and early-onset lone AF cases resides in constitutively expressed exons incorporated into the two principal cardiac isoforms[21]. The occurrence rate of TTNtv in control group B (frequency of TTNtv [PSI > 90] < 0.27) could be due to the fact that these individuals were free from cardiac diseases. All individuals had undergone cardiac examination and Holter monitoring without detecting any sign of arrhythmia. Furthermore, in accordance with our results, two recent GWAS have associated common *TTN* variants to AF susceptibility[5,25].

The clinical data among TTNtv carriers in the families suggest that LA dilation could be tachycardia induced (Table 3), as previously shown[26]. An essential feature of the patients belonging to the early-onset AF cohort is that they had all undergone extensive

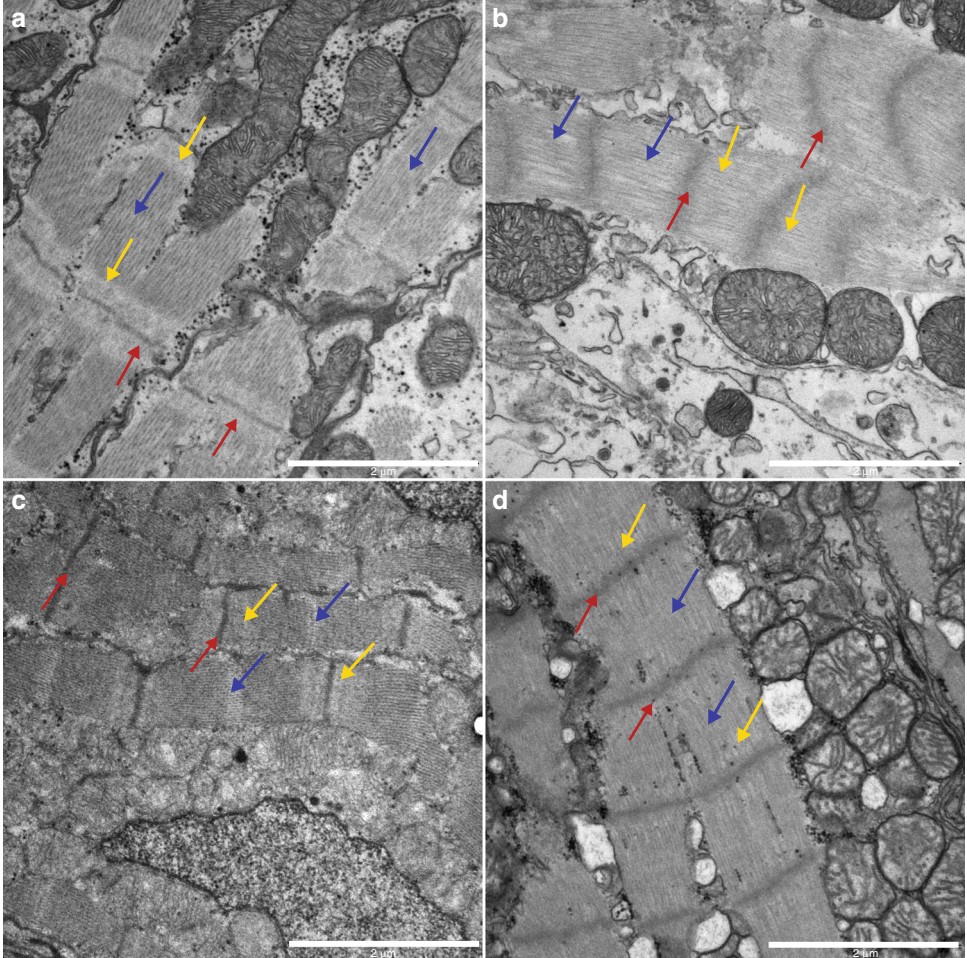

**Fig. 2** Compromised sarcomere structure in adult heterozygous zebrafish mutants. TEM images (13,500 × ) from adult atria (**a**, **b**) and ventricles (**c**, **d**). **a** WT atria show well-defined sarcomeres, with distinguishable Z-discs (red arrows) and I-bands (yellow arrows) throughout the tissue. Scale bar 2 μm. **b** In heterozygous mutant atria the sarcomere structure is less organized. The Z-discs appear blurred (red arrows), and the I-bands are absent (yellow arrows). Scale bar 2 μm. **c** Ventricle sarcomeres appear well defined in WT siblings, with clear Z-discs (red arrows), I-bands (yellow arrows), and M-lines (blue arrow). Scale bar; 2 μm. **d** In heterozygous mutants, there is a distinct lack of I-bands and M-lines, and the Z-discs appear blurry and increased in thickness (Scale bar; 2 μm). The length of the sarcomeres, as measured from Z-disc to Z-disc, are significant different between WT and heterozygous in atria

clinical screening, due to their diagnosis at such a young age. Yet, no known comorbidities were identified in these patients and none of the 18 early-onset lone AF TTNtv carriers have developed DCM after their AF diagnosis. Furthermore, echocardiographic examinations of up to 41 years after the AF diagnosis were normal. In this cohort, there is a ~10-fold increase in TTNtv, residing in constitutively expressed exons, compared with the general population. The clinical findings and the association of TTNtv to early-onset lone AF in our study suggest that TTNtv predisposes directly for AF.

Other cardiac structural genes have recently been associated with AF. Two different variants in the sarcomere protein gene myosin light chain 4 (*MYL4*) have in three independent studies been associated with familial early-onset AF[9,10,27]. Gudbjartsson et al.[10] performed whole-genome sequencing of the Icelandic population and identified a founder frameshift variant in *MYL4*, associated with a recessive form of AF. Orr et al.[9] reported a novel heterozygous *MYL4* variant in a family that in addition to early-onset AF also displayed signs of a primary atrial myopathy. When performing an overexpression of *MYL4* in zebrafish they observed both structural and electrical abnormalities associated with atrial cardiomyopathy and AF in humans.

Also, the protein product of the gene *SYNPO2L* which is expressed in the cardiac muscle, localized at the Z-disc and interacting with a number of other actin proteins, have been associated with AF[5,8]. The findings from the studies listed above support an emerging hypothesis of atrial cardiomyopathy as a mechanism that predisposes for AF.

From proteomics experiments, we confirmed that titin is highly expressed in the heart with no statistically significant difference between the protein expression level in the atria compared with the LV (Supplementary Fig. 8). The high expression level of titin in human atria suggests that TTNtv also should impact the atria. To investigate this, we used a CRISPR/Cas9 modified zebrafish carrying a N-terminal truncated variant of *ttn.2* (the cardiac-specific zebrafish orthologue of human *TTN*). We confirmed previous findings using immunostaining of isolated hearts from 72 h post fertilization (hpf) old *ttn.2*<sup>sfc9</sup> homozygous mutant and heterozygous fish. Homozygous zebrafish showed complete loss of the Z-discs in both atria and ventricle from an early larval stage, indicating a defective sarcomere structure in embryonic development and die around 5 days post fertilization (Supplementary Figs. 9–11). Heterozygous mutants appear phenotypically normal when evaluating the Z-discs in both atria and

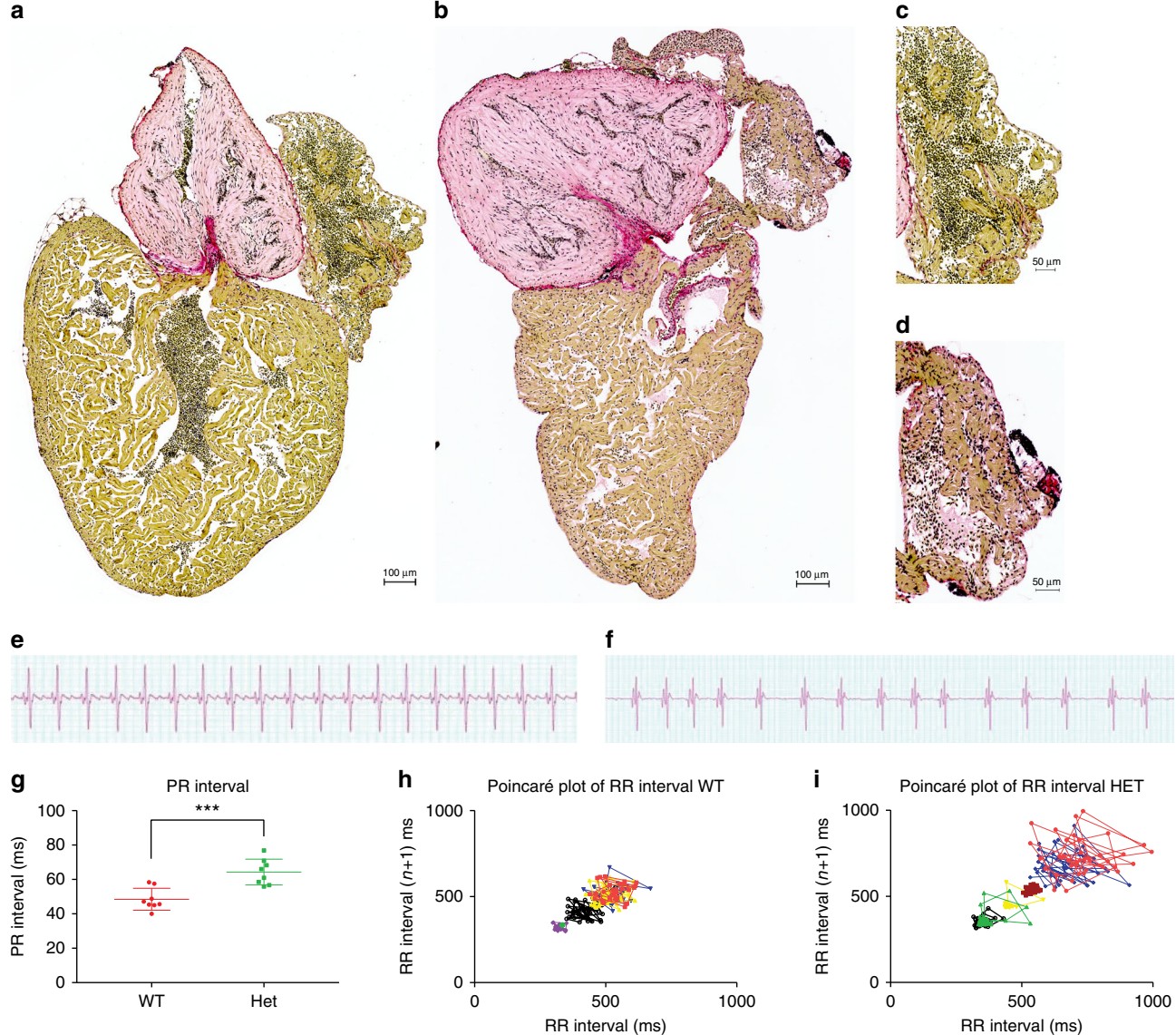

**Fig. 3** Truncating mutation in *ttn.2* cause increase fibrosis and electrophysiological defects in adult heterozygous mutants. Sirius staining of isolated whole hearts from adult WT (**a**) and heterozygous mutant siblings (**b**) show increased fibrotic lesions in the heterozygous heart (**b**) compared with the WT (**a**). This increase appears to be more pronounced in the atria of the heterozygous hearts (**d**) ($n = 4$) compared with that of the WT siblings (**c**) ($n = 4$). Scale bars: **a**, **b** 200 μm; **c**, **d**; 50 μm. ECG surface recordings of adult WT fish (**e**) revealed a regular ECG pattern with well-defined P-waves and QRS complexes, with regular PR intervals (**g**) and RR intervals, with low beat-to-beat variability, as shown by the Poincaré plot (**h**), indicative of a regular heart rhythm ($n = 8$). In the heterozygous siblings, the ECG pattern was equally well defined (**f**), but with a larger PR interval, compared with the WT siblings. Furthermore, the RR interval and beat-to-beat variability was irregular, as demonstrated by the Poincaré plot (**i**), indicating an irregular heart rhythm in the heterozygous siblings ($n = 8$)

ventricle. This is in accordance with previous reports on this mutant[19]. However, on closer inspection using TEM, the Z-discs of heterozygous mutants in the early larval stage appeared less defined as compared with WT siblings (Supplementary Fig. 12). Since our human TTNtv carriers are adults and heterozygous for the mutation, we focused on the heterozygous adult zebrafish mutant. TEM in adult heterozygous zebrafish revealed missing M- and I-lines in both atria and ventricle (Fig. 2). The sarcomere in the heterozygous adult hearts appeared disorganized with shorter sarcomere length and with unstructured Z-discs. Previous reports have investigated the presence of Z-disc as the hallmark of sarcomere integrity in titin truncations[19]. However, our data suggest that not only the Z-disc, but also the M-line and I-band are affected, since TEM imaging revealed a widespread sarcomere dysfunction in the atria.

TTNtv in cardiomyopathy patients have been shown to augment ventricular interstitial fibrosis and increase ventricular arrhythmias[28]. We speculated whether this mechanism could also be applied to the atria and cause AF. We therefore stained for fibrosis in adult hearts from WT zebrafish and their heterozygous siblings. The heterozygous *ttn.2^sfc9* adults did indeed have a higher degree of fibrosis in the atria, indicated by the Sirius Red staining (Fig. 3). Furthermore, ECG analyses showed electrophysiological defects with prolonged PR interval (Fig. 3, Supplementary Table 5). PR interval prolongation has recently been associated with AF[29].

Atrial fibrosis has continuously been reported to be more frequent in patients with AF compared with non-AF patients[30–33]. Subsequent studies have revealed that patients with atrial fibrosis have a high re-entrant activity, offering a likely explanation for

the increased burden of AF in these patients[34,35]. A few studies have included patients characterized as lone AF patients and even these patients display some amount of structural remodeling of the LA[36,37] with Mahnkopf et al. reporting that approximately one-fourth of the lone AF patients had moderate amounts of atrial fibrosis[37]. Based on these findings, Kottkamp[32] suggested to reclassify some lone AF patients as having "fibrotic atrial cardiomyopathy".

The finding of atrial fibrosis in the young adult zebrafish (~8 months of age) indicates that TTNtv predispose to the development of fibrosis in the atria from an early age. The early onset of AF in both investigated patient cohorts in the present study seems consistent with the zebrafish findings and provide a plausible link between structural disease and the electrical phenotype. However, we cannot exclude that other non-genetic factors, such as inflammation, might in concert with TTNtv cause an increased level of fibrosis.

According to current guidelines, catheter ablation of paroxysmal AF is a class I recommendation after failure of antiarrhythmic drug therapy[38]. If a patient's AF seems to result from atrial cardiomyopathy, it raises the question whether the patient could benefit from AF ablation, a dilemma recently discussed by Nattel and Dobrev[39] as AF patients with high amount of fibrosis have been shown to have a higher recurrence of post-ablation AF[35,40–42]. Interestingly, recent results from the CASTLE-AF trial[37] show a number of beneficial effects of catheter ablation in AF patients with associated heart failure. The ongoing DECAAF II study[43,44] investigating whether delayed enhancement magnetic resonance imaging-guided fibrosis ablation has better outcomes than traditional ablation techniques, might help address some of these issues. In conclusion, we found a significant enrichment of TTNtv in familial and early-onset lone AF. Using a zebrafish model, we showed that both larvae and adult TTN truncated fish have a compromised sarcomere structure and increased fibrosis in the atria. Furthermore, the heterozygous mutants also display an electrophysiological phenotype indicating a predisposition for arrhythmia. These data suggest that the atria are predisposed for disease independent of long-term systemic effects. Compromised sarcomere structure in the atria add a new dimension to the understanding of this common disease, as AF traditionally has been considered an electrophysiological disease.

## Methods

**Study subjects.** Familial AF cohort: Danish families with three or more family members diagnosed with AF were identified by the use of the Danish National Registries (The Civil Registration System, The National Patient Register, and The Danish Family Relations Database[16]). Danish citizens are assigned a unique personal identification number, which is registered in all contacts with public authorities enabling a complete survey of cases. AF was defined as a discharge diagnosis in the presence of the International Classification of Diseases (ICD)-8 code 427.93, 427.94, and ICD-10 code I48. The AF diagnosis obtained from the Danish National Patient Register has previously been proved valid, with a positive predictive value of 92.6%[45]. Individuals who agreed to participate had blood samples drawn, used for whole-exome sequencing, and clinical information was obtained through a questionnaire and review of patient health records. Written informed consent was obtained from all study participants. The study was approved by the scientific ethics committee for the Capital Region of Denmark (protocol number H-1-2011-044).

Early-onset lone AF cohort: A total of 399 unrelated early-onset AF subjects were recruited using the Danish National Patient Registries and ICD-10 code I48.9. Only patients with onset of disease before the age of 40 were included. Patients with cardiovascular diseases (mitral valve/aortic valve disease, ischemic heart disease, congestive heart failure, cerebral-vascular stroke, congenital heart malformation, and cardiomyopathy), or specific systemic diseases (hyperthyroidism, hypertension, and diabetes)[16] and/or with an abnormal echocardiography at the time of their AF diagnosis were excluded. Individuals fulfilling inclusion criteria were given written information and offered to participate. Blood samples and clinical data were collected. The study was approved by the scientific ethics committee for the Capital Region of Denmark (protocol number H-1-2011-044). All included patients gave written informed consent.

Control group A: The control population was drawn from an exome-sequenced subpopulation from the Inter99 study (CT00289237, ClinicalTrials.gov), previously described in detail by Glümer et al.[8] and Lohmueller et al.[14]. All participants were of self-reported Danish origin. This population was assumed to be representative of the general population (only AF affected individuals were excluded). From this population, we included all subjects who fulfilled our sequencing quality control criteria. The study was approved by the scientific ethics committee for the Capital Region of Denmark. All included patients gave written informed consent.

Control group B: Blood samples from 384 healthy men and women between 55 and 75 years of age and without a history of cardiovascular disease or stroke from the Copenhagen Holter Study[10] were used as controls. Informed consent was obtained from all participants. The study protocol was approved by the local ethics committee (KF 01 313322, KF 01 25304).

**DNA sequencing and genotyping.** Genomic DNA extracted from blood samples was used to construct DNA libraries. Whole-exome sequencing was performed on the familial AF cohort and control group A. In brief, ConGenomic DNA was extracted from peripheral blood using Maxwell 16 LEV Blood DNA Kit. The familial AF exomes were sequenced on Illumina HiSeq 2500 using Broad Institute (BI) bait capture kit (38 Mb target region). The control group A's exomes were sequenced with an Illumina HiSeq 2000 using Agilent SureSelect All Exon Kit v.2 (46 Mb target region), as previously described[15].

Targeted deep sequencing of *TTN* was performed on the lone AF cohort and control group B. The study subjects' DNA was fragmented by endonucleases and hybridized to biotinylated gene-specific probes incorporating Illumina paired-end sequencing motifs and indexed primers (Illumina TruSight cardio sequencing kit). The hybridized molecules were captured with magnetic beads, PCR amplified, and sequenced with the Illumina systems (HiSeq 2500 and NextSeq).

Raw reads were trimmed from adapter sequences and low-quality reads were filtered. Subsequently, alignment of reads were made to the to the human reference genome (NCBI Genome Build 37) with Burrow-Wheelers Aligner mem algorithm[46]. The alignments were post-processed according to Genome Analysis Toolkit version 3.4 (GATK) guidelines[47]. We performed a joint genotyping analysis on all exomes with Haplotypecaller/GATK v3.4 and on samples sequenced with targeted deep sequencing with Unifiedgenotyper/GATK v3.7.

**Evaluation of variants and samples in exomes.** Variant calling was performed with Haplotypecaller/GATKv.3.4 in the intersecting region of the two exome capture kits. The intersect region had 33 Mbp. Subsequently, variants were filtered using GATKv.3.4's tool VQSR with annotations:

$$-an\ QD - an\ FS - an\ SOR - an\ MQRankSum - an\ ReadPosRankSum$$

The filtering level threshold for sensitivity was set to 99.5% for SNPs and 99.0% for indels.

Hard filtering thresholds were set to:

$$QD<2.0, FS>200.0, MQ<40.0, ReadPosRankSum< -30.0$$

On a genotype level, calls with a genotype quality < 20 or read depth < 10 were discarded. Subsequently, variants with a missing rate > 20% or a Hardy–Weinberg equilibrium $P < 5 \times 10^{-6}$, calculated on family index patients and controls, were filtered.

Samples with a transition to transversion ratio (TiTv) or hetero-/homozygosity rate and singleton rate exceeding four SDs were excluded from further analyses.

**Evaluation of variants and samples in targeted sequencing.** In the targeted deep sequencing approach, variants were called with the software tool Unifiedgenotyper/GATK v3.7 in the protein-coding region of *TTN* that corresponds to the meta-transcript of ENSG00000155657, and a padding of 100 bp was added on ends of each exon region.

Hard filtering thresholds were set to:

$$QD<2.0, FS>60.0, MQ<40.0, ExcessHet>20.0,$$

$$MQRankSum< -20.0, ReadPosRankSum< -20.0$$

On a genotype level, calls with a genotype quality < 20 or read depth < 10 were discarded. Subsequently, any variants with a missing rate > 10% were excluded. Samples with a call rate < 95% were excluded. Samples with a TiTv ratio or hetero-/homozygosity rate and singleton rate exceeding four SDs were excluded.

The genetic distances were estimated with identity by state and MDS, using unrelated 1000 genomes project samples as reference (Supplementary Fig. 6). We considered samples exceeding 3 SD in any direction of the first two principal components given by the distribution from each respective group (lone AF cohort or control group B) and British in England and Scotland (GBR) populations an ethnic outlier. We applied the King robust algorithm with an LD threshold of 0.5, using R package SnpRelate[48], and excluded any duplicated or closely related samples (second degree or closer).

**Sequencing coverage exomes**. As a quality control measure, we filtered regions that on average had <70% covered with 10 reads or more in either cases or control groups. Subsequently, samples with mean or median < 35X or < 80% of targeted region covered with <10 reads, in any of the remaining autosome chromosomes, were excluded. Second, constitutively cardiac expressed exon regions in the *TTN* gene were classified as low coverage if < 80% were covered with <20 reads on average.

**Sequencing coverage targeted sequencing**. We defined the same minimum criterion for sequencing coverage in this replication study, i.e., constitutively cardiac expressed exon regions in the *TTN* gene were classified as low coverage and excluded if < 80% were covered with <20 reads on average.

**Inference of relatedness in families**. Reported family relatedness was inferred with the King robust algorithm on the filtered genotype dataset with an LD threshold of 0.5, using R package SnpRelate[48]. All reported family relationships were confirmed by the resulting kinship coefficients (Supplementary Fig. 4).

**Genetic relationship matrix**. The genetic relationship matrix (GRM) was based on the filtered dataset with an LD threshold of 0.5, excluding monomorphic- and non-autosomal variants and MAF > 0.01. We used AF family index patients and control samples to obtain the LD values. The GRM was generated using the methods PC-AiR and PC-relate available in R package GENESIS[49–51].

**Genetic analyses**. In the genetic analyses of familial AF exomes, we focused on rare LOF variants. Only family members with the clinical diagnosis of AF were taken into consideration, since 40–80% of all AF episodes are asymptomatic[17,18]. In a pair of monozygotic twins, one randomly chosen individual was excluded from the association test.

Rare variants were defined as non-common in dbSNP b.142, a MAF below 0.1% (or absent) in all of the ExAC and 1000 Genomes Project populations, and in an in-house Danish population of 2000 exomes[14,52–54]. Only variants with a putative high deleterious impact were included in the tests, as defined by the Sequence Ontology project[55]. Splice site region variants were evaluated with dbscSNV, where a conservative threshold, AdaBoost and Random Forests scores > 0.9, was set for splice-altering effects and classified as LOF variants[56].

For the genetic analyses of *TTN* in early-onset lone AF subjects, only rare LOF variants residing in constitutively expressed exons were considered, i.e., exons spliced into at least 90% of the *TTN* transcripts present in the heart (PSI ≥ 90).

**Statistical analyses**. We performed a Combined Multivariate and Collapsing (CMC) GLMM burden test of rare TTNtv residing in constitutively expressed exons[57] on the familial AF cohort and control group A. The GRM was generated using the methods PC-AiR and PC-relate on the filtered genotype dataset with an LD-pruning threshold of 0.5 and a minimum MAF of 1%[49,50]. The tests were performed using the statistical frameworks GENESIS and SeqVarTools[51,58] available as R packages. A binomial mixed model was first fitted under the null hypothesis using the generated GRM. The fitted mixed model was then used in conjunction to the burden test to adjust for polygenic random effects. A weight of one was given to all variants and only variants with internal MAF < 1% were included. The allele frequencies were calculated using AF family index patients and control samples. An exome-wide adjusted significance threshold $P < 2.5 \times 10^{-6}$ was considered significant.

With quality controlled and ethnically matched samples from the early-onset lone AF cohort and control group B, we tested the hypothesis of enrichment of rare TTNtv in AF cases. Only variants with internal MAF < 1% were included. The burden test was performed with Firth logistic regression and a $P < 0.05$ considered significant.

**Proteomics experiments on human heart tissue biopsies**. We used results from proteomics experiments to evaluate the protein expression level of titin in human atria compared with human ventricles. Briefly, three cardiac tissue biopsies from LA, right atria (RA), and LV were obtained from seven patients undergoing mitral valve surgery, and immediately snap frozen in liquid $NO_2$ and stored (–80 °C). Protein extraction of biopsies (2 mg per sample) was followed by digestion as described previously[59,60]. Desalted digested peptides were fractionated into 12 fractions by micro-flow reverse-phase ultrahigh pressure liquid chromatography. Following this, samples were separated on 15 cm fused-silica emitter columns (in a 1 h multi-step linear gradient). These were analyzed by online reversed-phase liquid chromatography coupled to a Q-Exactive Plus quadrupole Orbitrap tandem MS. Using the MaxQuant software, we processed the raw MS data. To identify proteins, we used a built-in Andromeda search engine containing human SwissProt protein entries. A total of 6588 proteins were measured of which 5941 were quantified. Titin was identified in all biopsies and quantitation was based on 2974 distinct peptides that in total cover 68.6% of the titin amino-acid sequence. One LA sample behaved as an outlier due to very limited input material available in that particular LA biopsy. The outlier was determined by number of protein identifications and principal component analysis and it was excluded from further

analyses. Rank analysis was based on summed MS-based protein intensities. Label-free quantification was performed in MaxQuant. Shapiro–Wilk normality test did not reject the null hypothesis that samples could be drawn from a normal distribution ($P > 0.05$ for all chambers), and Bartlett test of homogeneity of variances rejected the null hypothesis of equal variances between groups ($P = 0.04$). Protein intensities measured for titin in atrial and ventricular biopsies were compared using a two-sided Welch two sample t-test for samples of non-equal variance. Group means were not found to be statistically significantly different for any two cardiac chambers compared ($P = 0.19$ for RA versus LA, $P = 0.06$ for LA versus LV, and $P = 0.84$ for RA versus LV), nor for all atrial biopsies compared with all ventricular samples ($P = 0.54$). The experiment was not repeated.

Written informed consent was obtained from all study participants. The study was approved by the scientific ethics committee for the Capital Region of Denmark (protocol number 16238).

**Zebrafish model**. To evaluate the molecular consequences of a titin truncation, we used a CRISPR/Cas9 generated zebrafish[19]. The mutant *ttn.2^{sfc9}* carries a truncation at the N-terminal in the proximal I-band. The orthologous human amino-acid position (p.3048) corresponds to a similar localization as some of the identified TTNtv seen in our patients (Table 3, Supplementary Data 2). The mutant zebrafish was a kind gift from Rahul Deo, USCF, USA, and described in detail in Zou et al.[19]. The zebrafish carry a truncating mutation within the zebrafish cardiac *TTN* orthologue, *ttn.2*, so that the resulting protein product is truncated in the I-band.

**Histology and Sirius Red staining**. Adult zebrafish hearts were dissected from 8 months old adult heterozygous ($n = 4$) and WT siblings ($n = 4$). The tissue was fixed in 24 h at room temperature (RT) in 4% phosphate-buffered saline buffered formalin (Sigma-Aldrich, Denmark). After fixation, the samples were processed for paraffin embedding through dehydration in graded alcohols, cleared in xylene and embedded in paraffin. In all, 3 μm paraffin sections were cut and placed on glass slides. For Sirius Red staining, the slides were deparaffinized and rehydrated in xylene followed by graded alcohols. The staining was done using a standard Picosirius stain protocol.

**Immunofluorescence**. Hearts were isolated from 72 hpf old zebrafish as previously described[61], and stained for α-actinin (Sigma-Aldrich, mouse anti-α-actinin, clone EA-53, 1:500) and visualized using goat anti-mouse Alexa Fluor 488 secondary antibody (1:500; Molecular Probes, Invitrogen, DK)[61]. The hearts were imaged using a CorrSight (Fei, Germany) spinning disc microscope with an 40X EC Plan Neoflour (0.9NA). Images were post-processed using ImageJ and FigureJ software[62].

**Image analysis**. Following image capture, images were post-processed using Fiji/ImageJ, by applying a median filter, and background subtraction. Scale bars were applied using ImageJ[62].

**Transmitted electron microscopy**. Hearts from 6-month old WT ($n = 3$), 6-month old heterozygous siblings ($n = 3$), 72-h old WT ($n = 4$) and 72-h old WT heterozygous siblings ($n = 4$) were dissected out. Tissues from adults were separated into atrium and ventricle (12 samples in total). The samples were fixed in 2% v/v glutaraldehyde in 0.05 M sodium phosphate buffer (pH 7.4) for 24 h. Following fixation, samples were briefly rinsed in a sodium cacodylate buffer (0.15 M, pH 7.4). A 2 h post-fixation step followed (1% w/v OsO4, 0.05 M potassium ferricyanide in 0.12 M sodium cacodylate buffer, pH 7.4). Following the post-fixation incubation, samples were dehydrated in ethanol and transferred to propylene oxide, before embedding in Epon. Sections were cut at approximately 80 nm on a Leica UC7 microtome. Sections were collected on copper grids in Formvar supporting membranes, and stained with uranyl acetate and lead citrate. Samples were imaged on a Philips CM 100 TEM (Philips, Eindhoven, The Netherlands), operated at an accelerating voltage of 80 kV. Images were captured with an OSIS Veleta digital slow scan 2k × 2k CCD camera, and subsequently viewed in the ITEM software package.

**Sarcomere length analysis**. The length of the sarcomeres was analyzed using the TEM images and ImageJ. Using the Feret function in ImageJ, the number of pixels on the length between the two Z-discs were measured, and the length in μM was calculated using the scale bar from the microscope. The measured length of each sarcomere was plotted in Prism6 and analyzed with one-way analysis of variance and subsequent Tukey's test.

**Electrocardiography recordings on adult zebrafish**. All experiments were done in accordance with the European Convention for the Protection of Vertebrate Animals used for Experimental and other Scientific Purposes and the Danish Animal Experimental board (Personal project license 2017-15-0201-01305).

The zebrafish were anesthetized in 0.03% Tricaine/MS-222 (Sigma-Aldrich, St. Louis, Missouri, USA) in fish tank water, and after 5 min the zebrafish were placed dorsal side down in a damp sponge. The fish were orally perfused with fish tank water containing 0.03% Tricaine using a roller pump (ISM827B, ISMATEC

Germany) at 2 ml min$^{-1}$. Two custom made stainless steel electrodes were placed on top or slightly subcutaneously above the cardiac region using micromanipulators (Marzhauser MM33, Marzhauser Germany). The electrodes were connected to a differential AC amplifier (A-M Systems, WA, USA) with the following filter settings: Low cut-off filter = 10 Hz; high cut-off filter = 1 KHz; Gain = ×100. Signals were digitized in a PowerLab 4/30 (AD instruments, USA) and recorded at 10 k s$^{-1}$ in LabChart 7.0 (AD instruments, USA) using a digital low pass filter (cut-off frequency = 50 Hz; active input amplifier = 5 V range; low pass filter = 200 Hz; Mains filter = active). The ECGs were analyzed using minimum 30 consecutive beats, and analyzed beat for beat. The investigators had no knowledge of genotype status of the Zebrafish during ECG recording and measurement.

## Data availability

The datasets generated during analyses have been deposited at the European Genome-phenome Archive (EGA), which is hosted by EBI and CRG, under accession numbers EGAS00001003207 and EGAS00001003208. All relevant data are available upon request.

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

## Acknowledgements

We thank all patients for their participation. We thank Dr. Rahul Deo (UCSF, USA) for sharing the *ttn.2*-mutant zebrafish with us. We also thank the Zebrafish facility and staff at the University of Copenhagen for excellent zebrafish care. Furthermore, we thank CFIM for imaging assistance and TEM sample preparation. This work was supported by The John and Birthe Meyer Foundation, The Research Foundation at Rigshospitalet, Villadsen Family Foundation, The Arvid Nilsson Foundation, and The Hallas-Møller Emerging Investigator Novo Nordisk (NNF17OC0031204).

## Author contributions

Conceived and designed the research: M.S.O., A.G.H., M.M., M.F.R., G.A., P.R.L., L.R., S. P.O., and J.H.S. Data analysis and bioinformatics: G.A. Laboratory experiments: P.R.L. Performed statistical analysis: M.S.O., G.A., and S.R. Acquired the data: L.R., J.B.N., M.S. O., P.T.E., A.S., L.C., M.F.R., A.L., N.L., and P.R.L. Drafted the manuscript: L.R., M.S.O., G.A., L.A., and P.R.L. Made critical revision of the manuscript for key intellectual content: M.F.R., N.L., J.B.N., M.M., S.H., A.S., L.C., S.P.O., S.R., A.L., P.T.E., A.G.H., J.H.S.

## Additional information

**Competing interests:** The authors declare no competing interests.

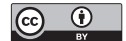

