## [Peer Review File · Nature Communications]

Reviewer #1 (Remarks to the Author):

The authors present data that implicates TTNtv variants as a risk for AF based both on familial and case-control cohort studies.

Major concerns:

1. The familial data is concerning and alone cannot be concluded to represent sufficient data implicating the TTNtv as the single gene cause for AF in these kindreds. These families have a very small number of affected individuals, no genetic information is provided for asymptomatic individuals (particularly important for older individuals), and an acceptable LOD score suggesting a single gene disease is not obtained. The possibility that AFIB in these kindreds is oligogenic is very real, with the TTNtv as a contributor.
2. Although TTNtv may increase risk of AF, based on the data provided, the role of sarcomeric genes/proteins contributing to AF is not at all novel. In the genetic disease of hypertrophic cardiomyopathy caused by sarcomeric gene mutations, up to 25% of cases have AFIB (Lee et al, Heart 2017).
3. On review of Pubmed, it is curious why the authors have not referenced the first paper implicating MYL4, an atrial sarcomeric gene, as a reported cause of AFIB that was published in 2016 (Orr et al, Nature Comm 2016) in the year prior to the Gudbjartsson et al. paper. In this paper there is considerable previously published discussion of AF as an atrial cardiomyopathy.
4. In reference to point 4 above, the Orr et al paper also used the ZF model and demonstrated a specific atrial myopathy related to the identified sarcomeric mutation.

Reviewer #2 (Remarks to the Author):

In this study by Refsgaard et al. the authors identify titin-truncating variants in 4/24 cases of familial atrial fibrillation. Compared to healthy controls they find that TTNtv are significantly enriched in patients with familial AF, and replicate this finding in an independent cohort of early-onset lone AF patients. Using CRISPR/Cas9 modified zebrafish carrying an N terminal TTNtv they observe defects in sarcomere assembly in early larval stage homozygous and adult heterozygous mutants, which they suggest supports a molecular predisposition for disease that begins in early cardiac development.

This is an interesting study that provides further support for the importance of myocardial structure in the pathogenesis of AF.

Major points:

- why did the authors select an N-terminal truncated variant for the zebrafish model? The AF patients have mutations across TTN according to Supplementary Figure 7. Did they test others?
- the authors refer to a study by Zou et al with regards to the mutant zebrafish line - there are no details on which allele was selected, was it just one? how the mutant was generated etc. there are minimal details in the Zou et al paper. There is also a lot of repetition e.g. page 17 line 377/378 essentially says the same as line 383/384. The methods also refer to supplementary figure 7 (line 386) but its not clear why, and that figure doesn't show the location of the CRISPR mutation.
- the authors state on Page 9 line 192 that to "assess the observed sarcomere defect in greater detail we performed TEM on isolated hearts from adult WT and heterozygous fish" - however above they state that in accordance with the Zou et al paper the heterozygous fish have normal z-discs,

and only the homozygous mutants have cardiac sarcomeric disarray - so why choose to carry out TEM in the adult hets only? Is it because the patients are hets? If so please clarify. If as the authors are suggesting, sarcomeric integrity is affected in early development do they see sarcomeric defects in the het fish in early larval stages?

- it would be important to further characterise the adult hets for defects in contractile function/cardiac conduction before making any conclusions about atrial cardiomyopathy predisposing for AF, I would expect the hets to have some sort of cardiac phenotype beyond that observed at the ultrastructural level based on the images in Figure 3.

Minor points:

- The authors state in the Results section - page 5 line 87/88 - that "Summary characteristics of the familial AF cohort are presented in Table 1" - however Table 1 only shows "Clinical information on the TTNtv and DSC2 loss-of-function carriers in the AF families".

- In Supplementary Table 2 they report "LOF variants with MAF < 10.000" - is this a typo (MAF is <0.1%)?

- There is a typo in the legend for Table 3 - "=proportion spliced in *in* the N2BA transcript"

- There is a typo on line 114 page 6 - the gene KCNE2 is mentioned twice.

- The reference numbering on pg 17 line 384 does not appear to be correct - Zou et al. is reference 18, not 14

Reviewer #3 (Remarks to the Author):

This is an interesting paper, I believe I may have reviewed it previously in an earlier form. I think the case for the role for TTNtv in AF is now well made with the discovery in the familial cases and the replication in early-onset cases. The fact that DCM does feature in the familial cases does raise the concern of the chicken-vs-egg problem of which of these diseases comes first, especially as TTNtv DCM is known to be associated with AF. But, I think the early-onset AF group - many who have had AF for a long time - who do not have features of DCM is a convincing cohort. Also, recently there have been a few GWASs showing association of AF with the TTN locus and this provides further substance to the authors claims. The TEM findings in the heterozygous TTNtv ZF (not the homozygous) are nice and actually this Reviewer observed similar aspects in a murine model that were never published. I would not show the homozygous TTNtv in the main figure as this does not model disease - just total sarcomere disruption. A recent study by the Heyman's group in EHJ showed that TTNtv in the ventricle was associated with fibrosis. Fibrosis is known to be important in AF - it would be nice to comment on a potential link between TTNtv, atrial fibrosis and AF as a potential unifying theory, if the authors felt so inclined.

Reviewer #4 (Remarks to the Author):

The authors present a nationwide Danish study on young patients suffering from lone atrial fibrillation and found that AF is associated with titin-truncating variants in a cohort without structural heart disease. Clinical follow-up and phenotyping of the analyzed patients to exclude further pathogenetic factors leading to AF was performed adequately.

Using the zebrafish as a disease model, Refsgaard et al. induced Titin-gene knockdown by CRISPR-Cas and found severe sarcomeric dysmorphology of both atrial and ventricular cardiomyocytes. Their findings are well illustrated, and the paper is likely to attract the interest of many scientists and to trigger further studies.

The study provides evidence in favour of the potential link between Titin truncation variants and atrial fibrillation. The authors use a very suitable animal model to evaluate their genetic findings in vertebrate model organism for numerous heart diseases. The authors demonstrate that Titin truncation induced by CRISPR-Cas leads to impaired assembly of the sarcomere in both atria and

ventricle at the early larval stage in homozygous fish and in adult heterozygous fish. There is a clinical perspective, as Titin truncation variants were observed GWAS studies.

Please address the following issues specifically:

1. Page 4, line 66:

Wrong! Several studies on AF ablation have shown strong clinical benefit for AF patients, regardless whether AF ablation is a causal therapy or not. Please revise this paragraph or differentiate more which therapeutic approach is ineffective (drug therapy, e.g.; STOP-AF trial). Recently, Marrouche and co-workers (CASTLE-AF Trial; NEJM 2017) could show a dramatic effect on mortality and morbidity for AF ablation in patients with heart failure.

2. Page 4, line 73:

Please describe in more detail the relation between AF and atrial cardiomyopathy (ac). Which aspect of ac (electrical, structural, etc.) could lead to AF? What is the mechanistic link between genetic variants in structural cardiac proteins and AF?

3. Page 4, 84ff:

Please comment in more detail why only 24 of 67 families could be included in the study. What was the most common reason for refusing informed consent?

4. Page 5, line 90ff:

Please mention the most common risk factors for AF. What is known about risk factors for AF in the youth? Can the others exclude subclinical viral myocarditis affecting the atrium? Are there any myocardial samples of affected patients available to evaluate for cardiac inflammation? If not, please address the interference of inflammation and structural defects (scar and fibrosis) in detail in the discussion.

5. Page 5, line 101:

What is the definition of AF the authors used? What is the difference of clinical AF and non-clinical AF? AF can be diagnosed also in patients without symptoms. However, in these patients 40-80% of AF episodes are not asymptomatic. Please revise this abstract in order to define more precisely inclusion and exclusion criteria of the study population.

6. Baseline characteristics:

What was the co-medication of the study group? How many patients underwent AF ablation? If patients were treated with drugs that are used for controlling heart rate or treat AF (β -blockers, digitalis, ACE inhibitors), please discuss how co-medication might influence clinical course of AF patients in regard to heart failure and DCM.

7. Page 11, line 234:

Indication for AF ablation is not a question of atrial cardiomyopathy. Symptomatic AF patients with paroxysmal non-valvular AF and failed drug therapy have a class-I-indication for catheter ablation according to the current guidelines. Please revise this passage or discuss this part more in detail.

8. Experimental approach:

What is known about AF in zebrafish in which Titin knock out was induced? Do hetero- or homozygous embryos or adult zebrafish demonstrate heart rhythm disorders? Since AF is still a chaotic activation of atrial cardiomyocytes it is also still an electrophysiological disease. Thus, I suggest performing electrophysiological studies on zebrafish such as ECG, voltage or calcium imaging or at least microscopic assessment of the underlying cardiac rhythm.

Reviewer 1.

Major concerns:

1. The familial data is concerning and alone cannot be concluded to represent sufficient data implicating the TTNtv as the single gene cause for AF in these kindreds. These families have a very small number of affected individuals, no genetic information is provided for asymptomatic individuals (particularly important for older individuals), and an acceptable LOD score suggesting a single gene disease is not obtained. The possibility that AFIB in these kindreds is oligogenic is very real, with the TTNtv as a contributor.

We thank the reviewer for this observant perspective. Atrial fibrillation (AF) is generally considered a complex disease and it is our view that TTNtv is a risk factor contributing to this complex/oligogenic trait. Regarding LOD score, our study was not designed for LOD score calculations and the identified families are not powered for such analyses (this would be more common in studies that look for monogenic causes). The maximum theoretical LOD score using parametric or non-parametric (combined LOD score) methods that can be obtained from these particular families would not be significant. However, AF has been shown to have a substantial heritable component. The most recent GWAS on AF (that we have co-authored) associated around 100 common loci with AF (Roselli et al., Nature Genetics 2018, in press), but these common loci only confer small odds ratio (OR) (1.1-1.9) and only explain a small fraction of the inheritance. In our study, we hypothesized that rare variants conferring a greater risk also contribute to the AF pathophysiology. Our hypothesis is that the probability of identifying high impact rare variants are greater in families with aggregation of AF and/or early onset age. Analyses of the whole exome sequencing data identified TTNtv in four independent families. We therefore performed an association test using mixed model logistic regression to see whether these were significantly enriched. Given the nature of the association test, we suggest to interpret these findings as an enrichment of TTNtv, conferring to a substantial risk for AF, even though we are not reporting on a Mendelian disease. In order to validate this claim, we have replicated our finding in an independent cohort, using logistic regression. We like to emphasize that the patients in this cohort are very young at onset of arrhythmia and do not have any comorbidities.

In order to clarify our viewpoint, we have added the following to the manuscript:

“This result suggests TTNtv as a major genetic contributor to AF, with a substantial risk for this complex oligogenic trait” (Page 10)

2. Although TTNtv may increase risk of AF, based on the data provided, the role of sarcomeric genes/proteins contributing to AF is not at all novel. In the genetic disease of hypertrophic cardiomyopathy caused by sarcomeric gene mutations, up to 25% of cases have AFIB (Lee et al, Heart 2017).

We thank the reviewer for this comment. We fully agree with the reviewer and Lee et al., 2017, that *hypertrophic cardiomyopathy* is epidemiologically recognized as a risk factor for developing AF. However, in our study we are not investigating the epidemiological or prognostic aspect but the genetic aspect. With regard to TTNtv, an important question that have arisen from studies such as Schafer et al. (NG, 2016) is whether this class of mutations are sole etiological factors causing dilated cardiomyopathy (DCM) or whether they ought to be considered as risk factors. This has some interesting implications in regard to treatment and prognosis that we hope future studies will address. None of the TTNtv positive patients in

our early-onset lone AF cohort have been diagnosed with DCM, even after long-term follow-up. We believe that the directional and mechanistic implication of TTNtv relationship to AF has not been shown before, and that this is quite novel compared to previous studies.

3. *On review of Pubmed, it is curious why the authors have not referenced the first paper implicating MYL4, an atrial sarcomeric gene, as a reported cause of AFIB that was published in 2016 (Orr et al, Nature Comm 2016) in the year prior to the Gudbjartsson et al. paper. In this paper there is considerable previously published discussion of AF as an atrial cardiomyopathy.*

4. *In reference to point 4 above, the Orr et al paper also used the ZF model and demonstrated a specific atrial myopathy related to the identified sarcomeric mutation.*

We thank the reviewer for pointing our attention to this. We have now included this reference: “Orr et al.¹⁷ reported a novel heterozygous *MYL4* variant in a family that in addition to early-onset AF also displayed signs of a primary atrial myopathy. When performing an overexpression of *MYL4* in zebrafish they observed both structural and electrical abnormalities associated with atrial cardiomyopathy and AF in humans.” (page 11).

Reviewer 2

Major points:

Why did the authors select an N-terminal truncated variant for the zebrafish model? The AF patients have mutations across TTN according to Supplementary Figure 7. Did they test others?

We thank the reviewer for this interesting comment. The TTNtv A-band mutant zebrafish display skeletal muscular defects, which are not seen in the I-band mutants (Zou et al., eLife 2015). To focus on the cardiac phenotype and the fact that two of our four familial mutations are located in the I-band, made us focus on this specific mutant. We have, so far, not investigated any other mutations.

We have added the following to the online method section:

“The mutant *ttn.2^{scf9}* carries a truncation at the N-terminal in the proximal I-band. The orthologous human amino acid position (p.3048) corresponds to a similar localization as some of the identified TTNtv seen in our patients.” (page 19)

The authors refer to a study by Zou et al with regards to the mutant zebrafish line - there are no details on which allele was selected, was it just one? how the mutant was generated etc. there are minimal details in the Zou et al paper. There is also a lot of repetition e.g. page 17 line 377/378 essentially says the same as line 383/384. The methods also refer to supplementary figure 7 (line 386) but its not clear why, and that figure doesn't show the location of the CRISPR mutation.

Thank you for this question. We contacted Dr. Rahul Deo to attain more details on the zebrafish line. In the CRISPR/Cas9 generated lines described in the Zou et al., 2015, eLife paper, there are six mutants described. Upon contact, Dr. Rahul Deo kindly shared mutant *ttn.2 N2 (ttn.2^{scf9})* with us. This mutant has a 1bp insertion, corresponding to the amino acid change A2825G in the zebrafish protein sequence, and

corresponding to the human orthologue position 3048, leading to a truncating variant of *ttn.2*. You are indeed correct that the reference to supplementary figure 7 is misplaced. We have deleted this reference. The repetition has also been corrected. Thank you for bringing this to our attention.

The authors state on Page 9 line 192 that to "assess the observed sarcomere defect in greater detail we performed TEM on isolated hearts from adult WT and heterozygous fish" - however above they state that in accordance with the Zou et al paper the heterozygous fish have normal z-discs, and only the homozygous mutants have cardiac sarcomeric disarray - so why choose to carry out TEM in the adult hets only? Is it because the patients are hets? If so please clarify. If as the authors are suggesting, sarcomeric integrity is affected in early development do they see sarcomeric defects in the het fish in early larval stages?

We apologize if the explanation behind the rationale for investigating only the heterozygous adults appeared unclear. It is correctly understood that we investigated the adult heterozygote zebrafish with TEM, because these resembles our patients which are adults and heterozygous carriers. When investigating the larvae zebrafish with immunofluorescence, only the homozygous fish have sarcomeric defects. The homozygous mutation in zebrafish is embryonic lethal, and the larvae die around 5 days post fertilization, making it impossible to study the adult homozygotes.

The presence of a sarcomere defect in adult heterozygous fish also support the pathogenic nature of these mutations.

We have made changes to the paragraph "zebrafish model" in the Results section (page 9) to clarify our approach.

"Zou et al. have previously shown that a CRISPR/Cas9 generated homozygous mutant carrying an N-terminal truncated variant of titin, *ttn.2^{sf919}*, displayed a severe sarcomere defect. The truncation is in proximity to one of the patient specific mutations identified in the current study, and this mutant zebrafish was used to further investigate the potential effects of TTNtv. The z-discs of WT, heterozygous, and homozygous larvae were stained with the z-disc protein, alpha-actinin and visualized with confocal microscopy (**Supplementary Figure 9-11**). We replicated the previous findings by Zou et al., where the WT and heterozygous larvae appeared with normal z-discs, whereas the homozygous mutants showed a severe sarcomere defect, with absent z-discs. The homozygous variant in zebrafish is embryonic lethal, and the larvae die around 5 days post fertilization. All patients carrying TTNtv in this study were heterozygous adults and we therefore decided to study the adult fish. To observe the sarcomere in greater detail we performed transmission electron microscopy (TEM) on isolated hearts from early larval stage (WT, homo- and heterozygous), adult heterozygous and WT fish. The TEM analysis revealed a sarcomere defect in both atria and ventricle of the adult and larval heterozygous mutants" (page 9)

Furthermore, we have now expanded the TEM studies to early larval stages heterozygote fish. These fish also seem to have a compromised sarcomere as they have poorly defined Z-disk compared to WT. Please see supplementary figure 12.

We have added the following to the manuscript:

"We confirmed previous findings using immunostaining of isolated hearts from 72 hours post fertilization

(hpf) old *ttn.2^{sc9}* homozygous mutant and heterozygous fish. Homozygous zebrafish showed complete loss of the z-discs in both atria and ventricle from an early larval stage, indicating a defective sarcomere structure in embryonic development and die around 5 days post fertilization (**Supplementary Figure 9-11**). Heterozygous mutants appear phenotypically normal when evaluating the z-discs in both atria and ventricle. This is in accordance with previous reports on this mutant¹⁹. However, on closer inspection using TEM, the z-discs of heterozygous mutants in the early larval stage appeared less defined as compared to WT siblings (**Supplementary Figure 12**). Since our human TTNtv carriers are adults and heterozygous for the mutation, we focused on the heterozygous adult zebrafish mutant. TEM in adult heterozygous zebrafish revealed missing M- and I-lines in both atria and ventricle (**Figure 2**). The sarcomere in the heterozygous adult hearts appeared disorganized with shorter sarcomere length and with unstructured z-discs." (Page 12)

It would be important to further characterise the adult hets for defects in contractile function/cardiac conduction before making any conclusions about atrial cardiomyopathy predisposing for AF, I would expect the hets to have some sort of cardiac phenotype beyond that observed at the ultrastructural level based on the images in Figure 3.

This a very good point, and we have now done ECG studies on the adult heterozygous zebrafish and their WT sibling. We have looked at p-wave duration, P-R interval, QRS interval, and RR interval. These data indicate that the adult heterozygous fish have a compromised electrical function, supported by significant prolonged PR intervals, please see Figure 3.

Minor points:

- *The authors state in the Results section - page 5 line 87/88 - that "Summary characteristics of the familial AF cohort are presented in Table 1", however Table 1 only shows "Clinical information on the TTNtv and DSC2 loss-of-function carriers in the AF families".*

Thank you for pointing this out. We have now corrected this.

- *In Supplementary Table 2 they report "LOF variants with MAF < 10.000" - is this a typo (MAF is <0.1%)?*

This is indeed a typo, thank you for pointing it out. It is supposed to say MAF < 1:10.000. This is now corrected.

- *There is a typo in the legend for Table 3 - "=percent spliced in *in* the N2BA transcript"*

Thank you again. We have deleted one of the "in".

- *There is a typo on line 114 page 6 - the gene KCNE2 is mentioned twice.*

You are correct. This has been corrected.

- *The reference numbering on pg 17 line 384 does not appear to be correct - Zou et al. is reference 18, not*

Thank you, this has been corrected.

Reviewer 3.

Reviewer #3 (Remarks to the Author):

The TEM findings in the heterozygous TTNtv ZF (not the homozygous) are nice and actually this Reviewer observed similar aspects in a murine model that were never published. I would not show the homozygous TTNtv in the main figure as this does not model disease just total sarcomere disruption. A recent study by the Heyman's group in EHJ showed that TTNtv in the ventricle was associated with fibrosis. Fibrosis is known to be important in AF it would be nice to comment on a potential link between TTNtv, atrial fibrosis and AF as a potential unifying theory, if the authors felt so inclined.

Thank you for this very encouraging review.

We have, as suggested, moved the figure showing the homozygous data to the supplementary appendix figure 11, and have instead included data from heterozygous larvae (TEM and staining). Please see new figures 2 and 3.

We agree that fibroses could explain the link between TTNtv and atrial arrhythmia.

In relation to this, we have conducted further investigations of the zebrafish. We investigated the amount of fibrosis in the hearts of the zebrafish using Sirius red staining, which revealed that the mutant zebrafish do indeed have more atrial fibrosis when compared to wildtype siblings. Please Figure 3 and results on page 9. Furthermore, we have added the following to the discussion:

“TTNtv in cardiomyopathy patients have been shown to augment ventricular interstitial fibrosis and increase ventricular arrhythmias²⁵. We speculated whether this mechanism could also be applied to the atria and cause AF. We therefore stained for fibrosis in adult hearts from WT zebrafish and their heterozygous siblings. The heterozygous *ttn.2^{sfc9}* adults did indeed have a higher degree of fibrosis in the atria, indicated by the Sirius Red staining” (page 12)

We have also conducted ECG analyses on the adult heterozygous zebrafish and the WT siblings, to assess the functional consequences of the mutations. These data suggest that fish with a truncating mutation in *ttn.2* have a compromised electrical cardiac function with significantly prolonged PR-intervals.

Reviewer 4.

(Remarks to the Author):

1. Page 4, line 66: Wrong! Several studies on AF ablation have shown strong clinical benefit for AF patients, regardless whether AF ablation is a causal therapy or not. Please revise this paragraph or differentiate more which therapeutic approach is ineffective (drug therapy, e.g.; STOPAF trial). Recently, Marrouche and coworkers (CASTLEAF Trial; NEJM 2017) could show a dramatic effect on mortality and morbidity for Af

ablation in patients with heart failure.

Thank you for pointing our attention to this very interesting paper. We have, as suggested, revised the paragraph regarding ablation and have included the Marrouche paper to the discussion.

“According to current guidelines, catheter ablation of paroxysmal AF is a class I recommendation after failure of antiarrhythmic drug therapy³³. If a patient’s AF seems to result from atrial cardiomyopathy, it raises the question whether the patient could benefit from AF ablation, a dilemma recently discussed by Nattel and Dobrev³⁴ as AF patients with high amount of fibrosis have been shown to have a higher recurrence of post-ablation AF^{30,35–37}. Interestingly, recent results from the CASTLE-AF trial³² show a number of beneficial effects following catheter ablation in AF patients with associated heart failure. The ongoing DECAAF II study^{38,39} investigating whether delayed enhancement MRI-guided fibrosis ablation have better outcomes than traditional ablation techniques, might help address some of these issues.” (page 13)

2. Page 4, line 73: Please describe in more detail the relation between AF and atrial cardiomyopathy (ac). Which aspect of ac (electrical, structural, etc.) could lead to AF? What is the mechanistic link between genetic variants in structural cardiac proteins and AF?

We thank the reviewer for this comment. We have conducted additional experiments. From our data, it appears that the atria in the heterozygous *ttn.2^{sf}c9* zebrafish have a higher degree of fibrosis, please see figure 3, and also a potential electrophysiological defect, demonstrated by prolonged PR-interval seen in an ECG analyses. This could be the mechanistic link between genetic variants in structural cardiac proteins and AF. These data have been added to the paper as an updated and we added this to the discussion:

“Furthermore, ECG analyses showed electrophysiological defects with prolonged PR interval, (**Figure 3, supplementary Table 6**). PR interval prolongation has recently been associated with AF²⁶.

Atrial fibrosis has continuously been reported to be more frequent in patients with AF compared to non-AF patients^{27–30}. Subsequent studies have revealed that patients with atrial fibrosis have a high re-entrant activity, offering a likely explanation for the increased burden of AF in these patients^{31,32}” (page 12)

3. Page 4, 84ff:

Please comment in more detail why only 24 of 67 families could be included in the study. What was the most common reason for refusing informed consent?

We agree that the inclusion of the families could be described in more detail and we therefore now added a more detailed paragraph about the inclusion to the supplementary appendix “Inclusion of families” page 8.

“In order for a family to be included, at least three members, from the same family, with the diagnosis of AF had to agree to participate. If only one or two family members with AF agreed to participate, the family was not included. In total, 24 families had at least three family members with an AF diagnosis, resulting in 77 AF cases.” (Supplementary Appendix page 7)

In accordance with the ethical approval, all patients can decline participation without any giving reason, which the majority of the patients did. Of the few who gave a reason for their decline, lack of time was the most prevalent reason.

4. Page 5, line 90ff:

Please mention the most common risk factors for AF. What is known about risk factors for AF in the youth? Can the others exclude subclinical viral myocarditis affecting the atrium? Are there any myocardial samples of affected patients available to evaluate for cardiac inflammation? If not, please address the interference of inflammation and structural defects (scar and fibrosis) in detail in the discussion.

Thank you for this interesting comment. We have now added risk factors for AF:

“399 unrelated early-onset AF subjects were recruited using the Danish national patient registries and ICD-10 code I48.9. Only patients with onset of disease before the age of 40 were included. Patients with cardiovascular diseases (mitral valve/aortic valve disease, ischemic heart disease, congestive heart failure, cerebral-vascular stroke, congenital heart malformation, and cardiomyopathy), or specific systemic diseases (hyperthyroidism, hypertension, and diabetes) and/or with an abnormal echocardiography at the time of their AF diagnosis were excluded.” (page 15)

None of the patients have been diagnosed with myocarditis (no diagnosis of such were described in the patient health records and none of the patients have reported present or previous disease record of myocarditis).

We found an increased amount of fibrosis in heterozygous zebrafish. Our data indicate that TTNtv might have a fibrogenic impact on cardiac tissue. Gene environment interaction would be extremely interesting to investigate further. We have added the following to the discussion:

“However, we cannot exclude that other non-genetic factors, such as inflammation, might in concert with TTNtv cause an increased level of fibrosis.” (Page 13)

5. Page 5, line 101:

What is the definition of AF the authors used? What is the difference of clinical AF and nonclinical AF? AF can be diagnosed also in patients without symptoms. However, in these patients 40-80% of AF episodes are not asymptomatic. Please revise this abstract in order to define more precisely inclusion and exclusion criteria of the study population.

Thank you for pointing this out, after reading it again we fully agree that our use of clinical AF is misguided. All patients included have been diagnosed with AF, whether or not they were asymptomatic. We have deleted the word clinical.

We have, in the Methods section, described how we identified the patients with the use of the Danish national registries (The Civil Registration System, The National Patient Register, and The Danish Family Relations Database). To be included in the study, the individuals must have a discharge diagnosis with the international classification of diseases (ICD)-8 code 427.93, 427.94, or ICD-10 code I48.

Only families, in which at least three family members with an AF diagnosis, and who agreed to participate were included into the familial cohort (please see further detailed description in answer to remark #3) We have updated the Methods section so the description of inclusion and exclusion details is more clearly outlined with regard to the early-onset lone AF cohort:

“399 unrelated early-onset AF subjects were recruited using Danish national patient registries and ICD-10 code I48.9. Only patients with onset of disease before the age of 40 were included. Patients with cardiovascular diseases (mitral valve/aortic valve disease, ischemic heart disease, congestive heart failure, cerebral-vascular stroke, congenital heart malformation, and cardiomyopathy), or specific systemic diseases (hyperthyroidism, hypertension, and diabetes) and/or with an abnormal echocardiography at the time of their AF diagnosis were excluded.” (page 15)

6. Baseline characteristics:

What was the comedication of the study group? How many patients underwent AF ablation? If patients were treated with drugs that are used for controlling heart rate or treat AF (β -blockers, digitalis, ACE inhibitors), please discuss how comedication might influence clinical course of AF patients in regard to heart failure and DCM.

Thank you for adding this interesting perspective. Two out of 12 familial AF patients with TTNtv and 8 out of eighteen lone AF patients had undergone ablation. We have added the ablation status of all the patients in Table 1 and 4.

The topic of comedication and a potential influence of the clinical course of AF is indeed very interesting. We have added Supplementary appendix table 3, with information regarding the families' medical treatment at inclusion. Six out of 15 received treatment for their AF with beta-blockers, digoxin, and/or ACE inhibitors. We have access to medical treatment at inclusion, and any long-term medical influence on the clinical course would be very interesting but speculative.

7. Page 11, line 234:

Indication for AF ablation is not a question of atrial cardiomyopathy. Symptomatic AF patients with paroxysmal nonvalvular AF and failed drug therapy have a class I indication for catheter ablation according to the current guidelines. Please revise this passage or discuss this part more in detail.

Thank you for this comment, we fully agree to make this clearer we have, as suggested, revised the paragraph, please see previous response to question 1. We also expanded our discussion on atrial disease with the following:

“A few studies have included patients characterized as lone AF patients and even these patients display some amount of structural remodeling of the LA^{33,34} with Mahnkopf *et al.* reporting that approximately one fourth of the lone AF patients had moderate amounts of atrial fibrosis³⁴. Based on these findings, Kottkamp²⁹ suggested to reclassify some lone AF patients as having “fibrotic atrial cardiomyopathy” (page 13)

8. Experimental approach:

What is known about AF in zebrafish in which Titin knock out was induced? Do hetero or homozygous embryos or adult zebrafish demonstrate heart rhythm disorders? Since AF is still a chaotic activation of atrial cardiomyocytes it is also still an electrophysiological disease. Thus, I suggest performing electrophysiological studies on zebrafish such as ECG, voltage or calcium imaging or at least microscopic assessment of the underlying cardiac rhythm.

We thank you for this very important comment. We have now conducted ECG studies on adult heterozygous zebrafish and WT siblings. These data show that the *ttn.2^{sf^{c9}}* fish has compromised electrical function as shown in the updated Figure 3, resembling characteristics of human AF; namely a longer PR-interval. These methods and results have now been added and the following to the discussion:

“We speculated whether this mechanism could also be applied to the atria and cause AF. We therefore stained for fibrosis in adult hearts from WT zebrafish and their heterozygous siblings. The heterozygous *ttn.2^{sf^{c9}}* adults did indeed have a higher degree of fibrosis in the atria, indicated by the Sirius Red staining (**Figure 3**). Furthermore, ECG analyses showed electrophysiological defects with prolonged PR interval, (**Figure 3, supplementary Table 6**). PR interval prolongation has recently been associated with AF²⁶.” (page 12)

Reviewer #1 (Remarks to the Author):

This manuscript aims to provide data supporting that TTNtv's contribute to AF etiology. The authors do this first by screening small kindreds with AF to identify TTNtv variants, and then screen a cohort of AF cases vs controls to show enrichment of rare variants. In addition, the authors present ZF data to imply mechanism of AF related to TTNtv's.

There remain major limitations and lack of novelty from this study.

1. First, the observed TTNtv's cannot be considered as definitive causes of AF in these kindreds, given the small family size and absence of genotype data in unaffected cases. At best, they may be contributing to AF.
2. Numerous studies have already implicated TTN as a contributor to AF, including an AF GWAS study (Christophersen, I. E. et al. 2017), and reports by Tayal et al (JACC 2017) and Hoorntje et al (EJHF 2017).
3. Previous studies have already demonstrated the effect of sarcomeric gene mutations on atrial sarcomeric structure (in rats and ZF), providing mechanistic insight into AF development (Peng W et al, JAHA 2017; Orr et al Nat Comm 2016).

As such, I struggle to find the novel information provided by this study.

Reviewer #2 (Remarks to the Author):

The authors have addressed my comments satisfactorily, and the manuscript is much improved.

Reviewer #4 (Remarks to the Author):

All of my comments have been Revised adaequately. Congratulation to the authors!

Point-by-point response to review 1.

1. First, the observed TTNtv's cannot be considered as definitive causes of AF in these kindreds, given the small family size and absence of genotype data in unaffected cases. At best, they may be contributing to AF.

We fully agree with the reviewer. We have added the following to the Results section “Genetic variation” (page 8):

“All variants co-segregated with disease, **however this does not imply a monogenetic cause.**”

We have also rephrased the discussion to make it more clear that we do not consider TTNtv as monogenetic causes of disease. This can be seen in the Discussion section (page 12):

“This result suggests TTNtv as a major genetic contributor to AF, **adding a substantial risk to this complex oligogenic trait.**”

2. Numerous studies have already implicated TTN as a contributor to AF, including an AF GWAS study (Christophersen, I. E. et al. 2017), and reports by Tayal et al (JACC 2017) and Hoorntje et al (EJHF 2017).

Thank you for this comment. These are indeed interesting studies in regard to our findings. Tayal et al. and Hoorntje et al. are now cited in our manuscript (Christophersen, I. E. et al. 2017 was already mentioned). We have also added the following sentence in conjugation to the citations:

“TTNtv have previously been strongly associated with cardiomyopathies²¹ **and some studies have reported an increased burden of arrhythmias among DCM patients with TTNtv^{22,23,}**

3. Previous studies have already demonstrated the effect of sarcomeric gene mutations on atrial sarcomeric structure (in rats and ZF), providing mechanistic insight into AF development (Peng W et al, JAHA 2017; Orr et al Nat Comm 2016).

The study by Peng W et al. 2017 have been added as a reference together with the other two

references that also associated *MYL4* with AF (Orr et al. Nat Comm 2016 and Gudbjartsson, D. F. et al. 2017, referenced in line 8 in the Introduction section).

We have adjusted the sentence from

“Two different variants in the sarcomere protein gene myosin light chain 4 (MYL4) have in two independent studies been associated with familial early-onset AF^{9,10,}”

to

“Two different variants in the sarcomere protein gene myosin light chain 4 (MYL4) have in three independent studies been associated with familial early-onset AF^{9,10,27}”